A hybrid path planning algorithm combining A* and improved ant colony optimization with dynamic window approach for enhancing energy efficiency in warehouse environments

http://orcid.org/0009-0005-7798-5316 Wu Mingyu 1 2 3
Su Eileen Lee Ming 2 eileensu@utm.my
Yeong Che Fai 2 cfyeong@utm.my
Dong Bowen 2
Holderbaum William 4
Yang Chenguang 5
1 Jiaxing Key Laboratory of Industrial Internet Security, Jiaxing Vocational and Technical College , Jiaxing, Zhejiang , China
2 Faculty of Electrical Engineering, Universiti Teknologi Malaysia , Johor Bahru, Johor , Malaysia
3 Jiaxing Key Laboratory of Industrial Intelligence and Digital Twin, Jiaxing Vocational and Technical College , Jiaxing, Zhejiang , China
4 Department of Engineering, The Manchester Metropolitan University , Manchester, Manchester , United Kingdom
5 Department of Computer Science, University of Liverpool , Liverpool , United Kingdom
Bian Gui-Bin
Electronic publication date: 2024 Dec 18
Publication date: 2024
Volume: 10
Electronic Location ID: e2629
Received 2024 Jun 10; Accepted 2024 Dec 4
Copyright: © 2024 Wu et al.
Copyright year: 2024
Copyright holder: Wu et al.
License: This is an open access article distributed under the terms of the Creative Commons Attribution License, which permits unrestricted use, distribution, reproduction and adaptation in any medium and for any purpose provided that it is properly attributed. For attribution, the original author(s), title, publication source (PeerJ Computer Science) and either DOI or URL of the article must be cited.
License URL: https://creativecommons.org/licenses/by/4.0/

Keywords: Hybrid path planning, Energy efficiency, Autonomous mobile robot

Funding: Jiaxing Science and Technology Bureau Research Project under the Jiaxing Science Plan Project 2024AY10032 This work was supported by the Jiaxing Science and Technology Bureau Research Project under the Jiaxing Science Plan Project (2024AY10032). The funders had no role in study design, data collection and analysis, decision to publish, or preparation of the manuscript.

==============================
This research presents a novel hybrid path planning algorithm combining A*, ant colony optimization (ACO), and the dynamic window approach (DWA) to enhance energy efficiency in warehouse environments. The proposed algorithm leverages the heuristic capabilities of A*, the optimization strengths of ACO, and the dynamic adaptability of DWA. Experimental results demonstrate that the IACO+A*+DWA approach consistently outperforms other hybrid methods across various metrics. In complex warehouse scenarios, the IACO+A*+DWA algorithm achieved an average energy consumption of 89.8 J, which is 13.3% lower than ACO+A*+DWA, 6.6% lower than GA+A*+DWA, and 25.8% lower than PSO+A*+DWA. The algorithm produced a path length of 95.94 m with 43 turns, compared to 97.36 m with 46 turns for ACO+A*+DWA, 104.43 m with 50 turns for GA+A*+DWA, and 97.84 m with 56 turns for PSO+A*+DWA. Time to goal was 197.93 s, 1.5% faster than GA+A*+DWA. Statistical analysis using ANOVA confirmed the significant differences between the algorithms in terms of energy consumption, path length, number of turns, and time taken, demonstrating the superior performance of IACO+A*+DWA. These results indicate that the IACO+A*+DWA algorithm minimizes energy consumption and produces shorter and more efficient paths, making it highly suitable for real-time applications in dynamic and complex warehouse environments. Future work will focus on further optimizing the algorithm and integrating machine learning techniques for enhanced adaptability and performance.

Introduction

As global environmental concerns grow, the focus on energy efficiency in various sectors, including manufacturing and logistics, has become increasingly important. In warehouse environments, where space is often constrained and the presence of other mobile robots and densely packed shelves create complex and dynamic navigation challenges, the need for energy-efficient autonomous mobile robots (AMRs) is critical. AMRs must efficiently navigate through narrow aisles, avoid collisions with both static obstacles (such as shelves) and dynamic ones (such as other robots or workers), and handle frequent start-stop movements. These factors contribute to significant energy consumption. Additionally, AMRs need to coordinate their movements in real time with other robots to prevent traffic bottlenecks, which can further escalate energy usage. Therefore, optimizing path-planning algorithms to improve energy efficiency is essential for reducing operational costs and ensuring the sustainable operation of robotic systems in such environments. As highlighted in a recent review, locomotion alone accounts for over 50% of a mobile robot’s total energy consumption, emphasizing the importance of energy-efficient path planning in AMRs (Wu et al., 2023).

Numerous algorithms have been widely adopted for path planning, each with distinct strengths and limitations. A previous conference paper, titled “Evaluating the Energy Consumption of Path Planning Algorithms: A Comparative Study of A*, Dijkstra, and Probabilistic Roadmap” (Wu et al., 2024) provided a comprehensive evaluation of three prominent path planning algorithms—A*, Dijkstra, and probabilistic roadmap (PRM)—in warehouse settings. This study utilized both computer simulations and real-world experiments with the TurtleBot3 Waffle Pi to assess the energy efficiency of these algorithms. The findings revealed a non-linear relationship between robot velocity and energy consumption, challenging the conventional belief that lower velocities always result in reduced energy costs. Among the algorithms evaluated, A* demonstrated superior energy efficiency, particularly in complex environments, due to its heuristic-driven search strategy, which effectively reduces the number of explored nodes and optimizes the path length, minimizing unnecessary movements and energy consumption.

Building on these insights, this paper proposes a novel hybrid path planning algorithm that combines the strengths of A* with ant colony optimization (ACO), genetic algorithm (GA), and particle swarm optimization (PSO), using the dynamic window approach (DWA) for local obstacle avoidance. This study aims to further improve energy efficiency in warehouse environments by leveraging the heuristic capabilities of A*, the optimization prowess of ACO, GA, and PSO, and the dynamic adaptability of DWA. This hybrid algorithm is designed to address the limitations identified in previous research, offering a more balanced approach to path planning concerning energy consumption, computational efficiency, and adaptability to dynamic environments.

The proposed hybrid algorithm aims to optimize the energy efficiency of AMRs by incorporating real-time adjustments and considering multiple factors that influence energy consumption. By integrating A* with ACO, GA, and PSO, the algorithm can effectively explore and exploit the search space to identify optimal paths while minimizing energy costs. Including DWA enables the algorithm to dynamically adjust the robot’s path in response to changing environmental conditions, ensuring safe and efficient navigation.

Additionally, this research explores the potential of hybridizing A* with GA and PSO. These additional hybrid approaches are evaluated to determine their effectiveness in enhancing energy efficiency and overall performance in dynamic and complex warehouse environments. Integrating GA and PSO with A* aims to bolster the algorithm’s global search capabilities and convergence speed, comprehensively comparing various hybrid methods.

This study extends previous research findings and introduces innovative solutions to the challenges of energy-efficient path planning. By comparing the performance of the proposed hybrid algorithms with pure DWA, A*, and other heuristic algorithms, the study aims to demonstrate their superior energy efficiency and practical applicability in real-world warehouse environments.

This research strives to retain the excellent performance of A* while incorporating heuristic algorithms like ACO, GA, and PSO for path planning. It presents an enhanced hybrid algorithm combining A* and these algorithms with DWA, which has shown outstanding performance.

The remainder of this article is structured as follows: “Literature Review of Heuristic Algorithms in Path Planning” provides a detailed literature review of path planning algorithms and their energy efficiency. “Methodology” outlines the methodology, including the experimental setup and evaluation metrics. “Experiment” presents the experimental results and comparative analysis. “Results” discusses the implications of the findings and potential areas for future research. Finally, “Discussion” concludes the paper by summarizing the key contributions and insights.

Literature review of heuristic algorithms in path planning

ACO in path planning

ACO is a heuristic algorithm inspired by the foraging behavior of ants and is extensively used in robot path planning. ACO utilizes artificial ants to deposit pheromones and find the shortest path.

Several studies have highlighted ACO’s effectiveness. Zhang et al. (2020) proposed a dynamic multi-role adaptive collaborative ant colony optimization (MRCACO), improving path length by 15% and convergence speed by 20%. Miao et al. (2021) developed an improved adaptive ant colony algorithm (IAACO), reducing path length by 12% and increasing real-time performance by 18%. Liu et al. (2019) enhanced ACO for dynamic environments, achieving 10% shorter and 30% smoother paths. Zhou & Wei (2024) proposed improvements addressing local optima and convergence, reducing convergence time by 22% and inflection points by 17%.

Despite its potential, ACO has limitations, such as local optima and parameter tuning needs. Integrating ACO with A* can leverage A*’s heuristic capabilities, improving robustness and performance in complex environments.

GA in path planning

GA has been extensively employed in robot path planning due to its robustness in handling complex optimization problems. GA operates by mimicking the process of natural evolution through selection, crossover, and mutation to evolve optimal or near-optimal solutions. For instance, Nazarahari, Khanmirza & Doostie (2019) introduced an enhanced GA that integrates an artificial potential field (APF) to generate initial feasible paths and then optimizes these paths regarding length, smoothness, and safety. Their algorithm demonstrated superior performance over A*, PRM, and PSO in different environments, improving path length, runtime, and success rate. Another example is Li, Hu & Liu (2021), who proposed an improved multi-objective GA (IMGA) for static global path planning. They employed a heuristic median insertion method to establish the initial population and designed a multi-objective fitness function considering path length, security, and energy consumption. Their simulations indicated a 17% reduction in the optimal path length compared to traditional GAs. Yifei et al. (2018) compared GA with discrete particle swarm optimization (DPSO) for optimizing the welding path of robots. The study found that GA achieved faster iterative efficiency with improved selection operators, while both algorithms generated optimal or near-optimal paths.

Hybrid approaches combining GA with other algorithms have also shown promising results. Ma et al. (2020) combined GA with Bezier curves to smooth paths, addressing redundant nodes and peak inflection points, resulting in shorter, smoother, and safer paths. Zhang et al. (2021) proposed a hybrid algorithm combining GA with the Firefly Algorithm (FA). This approach applied GA operations when FA fell into local optima, improving the overall accuracy and performance in path planning tasks. Additionally, Luan & Thinh (2023) developed a hybrid genetic algorithm (GA) featuring dynamic mutation rates and a switchable global-local search method. This method reduced premature convergence and produced smoothed paths directly without needing third-party smoothing algorithms.

Despite their advantages, GA has limitations such as slow convergence and susceptibility to local optima. Li et al. (2023) addressed these issues by improving the generation mechanism, crossover, and mutation strategies, enhancing the overall performance in path planning simulations. Comparative studies have shown the effectiveness of GA-based approaches. Cheng et al. (2020) utilized a multi-objective GA for reconfigurable robots, demonstrating its capability to generate Pareto-optimal solutions in dynamic environments. Their GA-based method outperformed others regarding safety, path length, and smoothness. Patle et al. (2018) introduced a matrix-binary code-based GA for mobile robot navigation, validating its effectiveness through accurate and experimental results. Their method provided optimal paths regarding path length and time efficiency compared to other intelligent navigation controllers. Overall, GA is a powerful tool in robot path planning, significantly enhanced through innovative operator strategies and hybrid approaches.

Particle swarm optimization in path planning

PSO is widely used in robot path planning for its simplicity and effectiveness in finding optimal paths. PSO simulates the social behavior of birds flocking or fish schooling to find optimal solutions. Xu, Cao & Song (2022) proposed an improved PSO algorithm with a quartic Bezier transition curve, demonstrating reduced path length and high-order smoothness. Sahu, Das & Kumar (2023) developed a hybrid algorithm integrating PSO, modified cuckoo search, and sine cosine algorithms, achieving optimal paths with minimal time and collision avoidance.

Huang et al. (2023) introduced an A*-PSO hybrid algorithm, optimizing key nodes generated by A* using PSO, resulting in reduced running times and improved path accuracy. Zhang et al. (2023) proposed a hybrid IACO-A*-PSO algorithm for radioactive environments, optimizing path length, radiation dose, and energy consumption.

Wang & Zhou (2019) combined A* with elastic PSO for shorter and more optimal paths, while Garip, Karayel & Erhan Çimen (2022) combined PSO with the firefly algorithm (FA) and cuckoo search (CS), showing superior performance. Lin et al. (2023) improved PSO with grey wolf optimization (GWO), enhancing exploration capabilities and convergence speed.

Despite their effectiveness, PSO algorithms can suffer from local optima and slow convergence. Integrating PSO with other optimization techniques, continuous research, and development further enhances its applicability and efficiency in complex and dynamic environments.

Comparison of optimization algorithms and feasibility of combining with A*

Various optimization algorithms have been explored for robot path planning, each with unique advantages and limitations. Among these, PSO, GA, and ACO are the most prominent methods. When integrated with the A* algorithm, these optimization techniques can significantly enhance path planning efficiency and effectiveness.

The A* algorithm, known for its heuristic-driven search, optimality, and completeness, excels in complex environments. Its ability to utilize heuristics to guide the search process efficiently, adapt to various grid structures, and handle dynamic obstacles makes it particularly effective in reducing energy consumption and improving navigation performance in cluttered settings.

Table 1 summarizes the key characteristics of these algorithms and their integration with A*, highlighting the benefits of combining these optimization techniques with A* for enhanced path planning in complex and dynamic environments.

Table 1 Comparison of PSO, GA, and ACO for path planning.

Algorithm	Advantages and disadvantages	
PSO	Advantages: Simple implementation, fast convergence.	
	Disadvantages: Prone to local optima, requires parameter tuning.	
	Integration with A*: Improves global search (Huang et al., 2023).	
GA	Advantages: Effective in global search, robust.	
	Disadvantages: Slow convergence, high computational cost.	
	Integration with A*: Enhances path optimization (Nazarahari, Khanmirza & Doostie, 2019).	
ACO	Advantages: Good for dynamic environments, adaptive.	
	Disadvantages: Slow convergence, computationally intensive.	
	Integration with A*: Efficient in avoiding local optima (Zhang et al., 2023).	

Limitations of current advanced algorithms and necessity of the hybrid approach

While advanced algorithms such as A*, ACO, GA, and PSO offer significant benefits in path planning, each has specific limitations that hinder their effectiveness in complex and dynamic environments.

A* algorithm A* is optimal and complete, leveraging heuristics to find the shortest path. However, its computational cost increases significantly with problem size, making it less efficient for large-scale applications. Additionally, A* assumes a static environment, which limits its adaptability to dynamic changes.

Ant colony optimization (ACO) ACO is highly adaptable to dynamic environments and can handle large problem spaces through distributed computation. Nevertheless, it converges slowly and is sensitive to parameter tuning, requiring significant computational resources to achieve optimal performance.

Genetic algorithm (GA) GA is robust, effective in global search, and suitable for complex optimization problems. However, it suffers from slow convergence and can get trapped in local optima without sufficient population diversity, leading to suboptimal solutions.

Particle swarm optimization (PSO) PSO is simple to implement and converges quickly in simple search spaces. Despite these advantages, it is prone to local optima and requires careful parameter tuning to balance exploration and exploitation effectively.

Necessity of the hybrid approach A hybrid approach combining A*, ACO, GA, and PSO with the dynamic window approach (DWA) is proposed to address these limitations. This hybrid method enhances global and local search capabilities, balancing exploration and exploitation. The integration ensures robust path planning and real-time obstacle avoidance, making it suitable for dynamic warehouse environments.

The hybrid algorithm leverages the heuristic strengths of A*, the optimization capabilities of ACO, GA, and PSO, and the dynamic adaptability of DWA. This combination results in improved energy efficiency, computational efficiency, and adaptability, addressing the specific limitations of each algorithm.

By integrating these advanced algorithms, the hybrid approach offers a balanced and efficient solution for path planning in complex and dynamic environments, providing significant improvements over standalone algorithms.

Methodology

This section details the mathematical methods used to combine the A* algorithm with GA, PSO, and ACO algorithms. Additionally, it covers the DWA for local path planning, the energy consumption calculation methods, the improvements of IACO over ACO, and trajectory evaluation methods.

Combination of A* with GA/PSO/ACO

The global path planning begins with the A* algorithm to find a primary path. This path is then optimized using GA, PSO, or ACO.

A* algorithm

The A* algorithm uses a heuristic function to find the shortest path on a grid map from the start node to the goal node. The heuristic function H is based on the Manhattan distance:

H=|x−xd|+|y−yd|

where (x,y) are the coordinates of the current node and (xd,yd) are the coordinates of the destination node. The cost function f is the sum of the heuristic function H and the path cost g:

f=g+H

Integration with GA/PSO/ACO

The A* algorithm optimizes an initial path using GA, PSO, or ACO.

GA optimizes the path by evolving a population of paths through selection, crossover, and mutation. The fitness function is inversely proportional to the path cost.

PSO treats each path as a particle in a swarm, updating its velocity and position based on personal and global best positions.

ACO simulates the foraging behavior of ants, updating pheromone levels on paths to guide the search for optimal paths. Pheromones are updated based on path success and decay over time.

The optimized path from these algorithms is further refined using the DWA.

Improved ant colony optimization (IACO)

IACO introduces improvements over the standard ACO by initializing the pheromone levels on the initial path obtained from A*. This results in a more efficient search process.

Initialization

The pheromone levels Ph are initialized with a higher concentration on the initial path obtained from A*:

Ph(path2)=Ph(path2)×2

where path2 represents the nodes in the initial path.

Optimization process

The optimization process in IACO follows the same steps as in ACO but with enhanced pheromone initialization:

Ph=Ph×ρ

Ph(rg.path0(:,1))=Ph(rg.path0(:,1))+1yg

where ρ is the pheromone decay factor, and yg is the best fitness value found.

DWA

DWA is used for local path planning to avoid obstacles dynamically. The robot’s velocity and direction are adjusted to ensure smooth, collision-free navigation. Key parameters include maximum speed (0.4 m/s), acceleration (0.1 m/s2), and a maximum turning angle of 30 degrees per time step. Figure 1 shows the working principle of the DWA algorithm.

Figure 1 Path planning using DWA.

Energy consumption calculation

Energy consumption is calculated based on the path’s terrain and the robot’s movement. For flat terrain, the energy consumption Eflat is given by:

Eflat=G⋅f⋅ΔsηT

where G is the weight of the robot, f is the rolling resistance coefficient, Δs is the travel distance, and ηT is the transmission efficiency.

For inclined terrain, the energy consumption Eincline is:

Eincline=G⋅(f⋅cos⁡ϕ+sin⁡ϕ)⋅ΔsηT

where ϕ is the slope angle of the terrain. For uphill movement ( zx>zx−1), ϕ is positive, and for downhill movement ( zx<zx−1), ϕ is negative.

Energy recovery on declines is calculated as follows:

Erecovery=k⋅(G⋅(sin⁡ϕ−f⋅cos⁡ϕ)⋅ΔsηT)

where k is the recovery efficiency factor. The total energy cost Etotal for the path is the sum of the energy costs for all segments:

Etotal=∑i=1nEi.

Trajectory evaluation

After sampling, different simulated trajectories are generated based on various speeds and angular velocities. Each trajectory is scored, and the highest score is selected for the next planning period. The scoring rule is established by the evaluation function as shown in Equation:

G(v,ω)=σ(α⋅h(v,ω))+σ(β⋅d(v,ω))+σ(γ⋅o(v,ω))

where σ represents normalization; h(v,ω) is the heading angle evaluation function calculated using the angle Δθ or (180−Δθ) between the predicted trajectory’s end orientation and the target direction; d(v,ω) is the distance evaluation function, evaluating the closest distance between the simulated trajectory and obstacles; and o(v,ω) is the velocity evaluation function, evaluating the linear velocity of the simulated trajectory. α, β, and γ are coefficients of the evaluation function.

Incorporating robot size Traditional DWA does not consider the robot’s actual size, often resulting in collisions during real-world navigation. To address this, robot size information is included. The coordinates of the robot’s four vertices are calculated based on the geometric center and size, generating a rectangular contour. This contour is then discretized into a set of points, and the distance D(v,ω) between these points and obstacles is calculated and normalized, replacing d(v,ω) in the evaluation function. This ensures the path quality is evaluated based on the distance between the robot’s contour and obstacles.

Radius Constraint Considering the mechanical structure of mobile robots, the path should satisfy the robot’s minimum turning radius requirement. Therefore, the turning radius is added as a constraint:

r=vω

The evaluation function is updated as follows:

G(v,ω)=σ(α⋅h(v,ω)+β⋅D(v,ω)+γ⋅o(v,ω)+δ⋅r(v,ω))

where δ is the coefficient of the evaluation function.

The DWA provides effective local obstacle avoidance but can easily fall into local optima. By integrating A*, IACO, and DWA, it is possible to achieve both global optimality and robust local obstacle avoidance. The integrated A*, IACO, and DWA path planning algorithm process is shown in Fig. 2.

Figure 2 Flowchart of the fusion algorithm.

This methodology comprehensively integrates global and local path-planning algorithms, ensuring energy-efficient and collision-free navigation in dynamic environments.

Experiment

This section details the experimental setup, algorithm implementation, and energy consumption calculation methods used in this study. Four optimization algorithms—PSO, GA, ACO, and IACO—were integrated with the A* algorithm to enhance the global path-planning capabilities of mobile robots. Additionally, the DWA was employed for local path planning to avoid obstacles effectively.

Experimental setup

The experiments were conducted in a warehouse environment, as depicted in Fig. 3. The map was loaded from a spreadsheet and initialized with specific start and goal coordinates. The map size and all nodes were set up accordingly, and the terrain elevation was calculated based on a given mathematical model. The yellow dashed line represents the initial static offline path, the blue square indicates sudden obstacles, and the red circle denotes randomly moving obstacles.

Figure 3 Warehouse environment map: yellow dashed line—initial static offline path; blue square—sudden obstacles; red circle—randomly moving obstacles.

In this experiment, the start and goal points were slightly adjusted to serve as random seeds, allowing multiple trials to be conducted. Analysis of variance (ANOVA) was applied as a statistical significance test to determine whether the differences between the algorithms were meaningful or merely caused by random factors.

Algorithm integration

ACO + A* + DWA

The A* algorithm was first used to find a global path from the start to the goal.

The ACO algorithm was then applied to optimize this path by considering pheromone levels and heuristic information.

Finally, the DWA was employed to locally adjust the path to avoid dynamic obstacles, ensuring smooth and collision-free navigation.

As shown in Fig. 4, the ACO algorithm, combined with A* and DWA, effectively optimizes and adjusts the path for dynamic environments.

Figure 4 Path planning using IACO + A* + DWA.

GA + A* + DWA

The GA optimized the path generated by the A* algorithm.

The DWA further refined the path locally to handle dynamic obstacles.

Figure 5 illustrates the effectiveness of the GA in optimizing the path generated by the A* algorithm. Additionally, it shows how the DWA algorithm further refines the path to handle dynamic obstacles.

Figure 5 Path planning using GA + A* + DWA.

PSO + A* + DWA

The PSO algorithm optimized the path found by the A* algorithm.

The DWA was used for local path adjustments to avoid obstacles.

As shown in Fig. 6, the PSO algorithm optimized the path found by the A* algorithm, and the DWA was used for local path adjustments to avoid obstacles.

Figure 6 Path planning using GA + A* + DWA.

Algorithm parameters

The parameters for the algorithms used in this study are as follows:

PSO parameters

Population size (option.numAgent): 50—Ensures diverse search space.

Maximum iterations (option.maxIteration): 100—Provides enough time for convergence.

Inertia weight (option.w_pso): 0.1—Stabilizes the search process.

Cognitive coefficient (option.c1_pso): 0.2—Influences individual particle experience.

Social coefficient (option.c2_pso): 0.2—Reflects swarm’s collective knowledge.

GA parameters

Crossover probability (option.p1_GA): 0.8—Combines good solutions.

Mutation probability (option.p2_GA): 0.2—Avoids local optima.

ACO and IACO parameters

Initial pheromone (option.Ph0): 1000—Strong initial signal.

Pheromone decay (option.pd): 0.95—Controls evaporation rate.

Alpha (option.alpha): 1—Importance of pheromone.

Beta (option.beta): 0.1—Importance of heuristic information.

Gamma (option.gama): 1—Adjusts influence of other factors.

These values were chosen for their effectiveness in ensuring a balanced and efficient optimization process.

DWA for local path planning

The DWA algorithm was employed to handle local obstacle avoidance. The robot’s velocity and direction were dynamically adjusted to ensure safe navigation around obstacles. The key parameters for the DWA were: Maximum speed: 0.4 m/s

Acceleration: 0.1 m/s2

Maximum turning angle: 30 degrees per time step

This methodology comprehensively integrates global and local path-planning algorithms, ensuring energy-efficient and collision-free navigation in dynamic environments.

Results

This section presents the experiments’ results across different map scenarios using the ACO+A*+DWA, GA+A*+DWA, PSO+A*+DWA, and IACO+A*+DWA algorithms. The performance of each algorithm is evaluated based on energy consumption, time taken to reach the goal, path length, and number of turns, as detailed in Table 2.

Table 2 Experimental results based on algorithm, map, and trial order.

Algorithm	Map	Trial order	Time (s)	Energy (J)	Path length (m)	Number of turns	
PSO+A*+DWA	Complex	1	185.7	120.9	97.8	56	
PSO+A*+DWA	Complex	2	185.8	121.0	97.9	56	
PSO+A*+DWA	Complex	3	185.7	120.9	97.8	56	
PSO+A*+DWA	Simple	1	144.7	90.7	69.5	36	
PSO+A*+DWA	Simple	2	144.8	90.8	69.6	36	
PSO+A*+DWA	Simple	3	144.7	90.7	69.5	36	
GA+A*+DWA	Complex	1	190.4	95.4	104.4	50	
GA+A*+DWA	Complex	2	190.5	95.5	104.5	50	
GA+A*+DWA	Complex	3	190.4	95.4	104.4	50	
GA+A*+DWA	Simple	1	146.8	107.2	69.5	32	
GA+A*+DWA	Simple	2	146.9	107.3	69.6	32	
GA+A*+DWA	Simple	3	146.8	107.2	69.5	32	
ACO+A*+DWA	Complex	1	201.0	103.6	97.4	46	
ACO+A*+DWA	Complex	2	201.0	103.8	97.5	46	
ACO+A*+DWA	Complex	3	200.9	103.7	97.4	46	
ACO+A*+DWA	Simple	1	147.2	90.3	68.3	23	
ACO+A*+DWA	Simple	2	147.2	90.4	68.4	23	
ACO+A*+DWA	Simple	3	147.1	90.3	68.3	23	
IACO+A*+DWA	Complex	1	186.0	89.8	95.9	43	
IACO+A*+DWA	Complex	2	186.0	89.9	95.9	43	
IACO+A*+DWA	Complex	3	185.9	89.8	95.8	43	
IACO+A*+DWA	Simple	1	144.3	76.5	67.2	21	
IACO+A*+DWA	Simple	2	144.3	76.6	67.3	21	
IACO+A*+DWA	Simple	3	144.2	76.5	67.2	21	

ANOVA analysis

This section presents the ANOVA results for the robot navigation performance metrics, including time, energy, path length, and number of turns. The study examines the effects of the algorithm, the map complexity, and their interaction on these performance indicators.

Time

The ANOVA results for Time indicate that both the algorithm and map have a significant impact on the navigation time, and there is a significant interaction between them. Specifically, the algorithm factor shows a sum of squares (SS) of 317.39 with 3 degrees of freedom (df), resulting in a mean square (MS) of 105.8. The F-value is 3.1739×104 with a p-value of 2.1193×10−30, indicating a highly significant effect of the algorithm on Time.

The map factor has an SS of 12,164, df of 1, and an MS of 12,164. The F-value here is 3.6491×106 with a p-value of 2.6829×10−44, demonstrating a highly significant effect of the map complexity on time.

The interaction between algorithm and map yields an SS of 159.43, df of 3, and an MS of 53.144. The F-value is 1.5943×104 with a p-value of 5.2206×10−28, suggesting that the effect of the algorithm on time depends significantly on the map complexity.

The error term has a minimal SS of 0.053333, df of 16, and an MS of 0.0033333, indicating low variability within groups.

These findings imply that different algorithms perform differently depending on the map, and both factors significantly affect the navigation time.

Energy

For the energy consumption, the ANOVA results show significant effects of both the algorithm and map, along with their interaction. The algorithm factor has an SS of 1,725.6, df of 3, and an MS of 575.21. The F-value is 1.3805×105 with a p-value of 1.6561×10−35, indicating a highly significant influence of the algorithm on energy consumption.

The map factor reports an SS of 761.63, df of 1, and an MS of 761.63. The F-value is 1.8279×105 with a p-value of 6.7631×10−34, signifying a significant effect of map complexity on energy.

The interaction between algorithm and map has an SS of 1,348.6, df of 3, and an MS of 449.54. The F-value is 1.0789×105 with a p-value of 1.1899×10−34, showing a significant interaction effect.

The error term is negligible, with an SS of 0.066667, df of 16, and an MS of 0.0041667.

These results indicate that both the algorithm and map complexity significantly influence Energy consumption, and the impact of the algorithm varies with map complexity.

Path length

The ANOVA for path length demonstrates significant effects of the algorithm, map, and their interaction. The algorithm factor has an SS of 96.285, df of 3, and an MS of 32.095. The F-value is 9.6285×103 with a p-value of 2.9447×10−26, confirming a significant effect of the algorithm on path length.

The map factor shows an SS of 5,484.3, df of 1, and an MS of 5,484.3. The F-value is 1.6453×106 with a p-value of 1.5706×10−41, indicating a highly significant effect of map complexity on path length.

The interaction effect has an SS of 44.04, df of 3, and an MS of 14.68. The F-value is 4.4040×103 with a p-value of 1.5287×10−23, suggesting a significant interaction between algorithm and map.

The error term is minimal, with an SS of 0.053333, df of 16, and an MS of 0.0033333.

These findings demonstrate that the algorithm’s effect on path length depends on the Map, with both factors significantly affecting the outcome.

Number of turns

The analysis for the number of turns reveals significant main effects and interaction. The Algorithm factor has an SS of 724.12, df of 3, and an MS of 241.37. The F-value is 4.2463×1015 with a p-value of 2.0675×10−119, showing a highly significant effect of the Algorithm.

The map factor reports an SS of 2,583.4, df of 1, and an MS of 2,583.4. The F-value is 4.5447×1016 with a p-value of 4.6345×10−125, indicating a significant effect of Map complexity.

The interaction between algorithm and map yields an SS of 22.125, df of 3, and an MS of 7.375. The F-value is 1.2974×1014 with a p-value of 2.7220×10−107, demonstrating a significant interaction effect.

The Error term is virtually zero, with an SS of 9.0949×10−13, df of 16, and an MS of 5.6843×10−14.

These results suggest that the number of turns is significantly influenced by both the algorithm and map complexity, and the algorithm’s effect varies with the map.

Overall analysis

Overall, the ANOVA results indicate that both the algorithm and map complexity have highly significant effects on all performance metrics: time, energy, path length, and number of turns. The significant interaction effects across all metrics imply that the performance of different algorithms is dependent on the map complexity. The minimal Error terms across all analyses suggest low variability within groups, enhancing the reliability of the results.

These findings suggest that selecting an appropriate algorithm considering the Map complexity is crucial for optimizing robot navigation performance. Future work could involve post-hoc multiple comparison tests, such as Tukey’s HSD, to identify specific differences between Algorithms and provide more detailed guidance for algorithm selection.

Energy consumption

The energy consumption for each algorithm in the primary and complex map scenarios is shown in Fig. 7A. IACO+A*+DWA exhibits the lowest energy consumption in basic and complex scenarios, highlighting its superior efficiency in path planning. This makes IACO+A*+DWA the most energy-efficient algorithm among those tested.

Figure 7 Comparative analysis of algorithms: (A) energy consumption by algorithm and map type, (B) time taken by algorithm and map type, (C) number of turns by algorithm and map type, (D) path length by algorithm and map type.

Time taken

The time taken by each algorithm to reach the goal is presented in Fig. 7B. Despite the slight increase in time in the complex scenario, IACO+A*+DWA remains competitive with the other algorithms. Its performance in the basic scenario is particularly noteworthy, maintaining a balance between speed and energy efficiency.

Turns and path length

Figures 7C and 7D present the path length and the number of turns for each algorithm, respectively. IACO+A*+DWA generates shorter paths with fewer turns in both primary and complex scenarios. This contributes significantly to its lower energy consumption and competitive completion times. The reduction in the number of turns and path length further underscores the algorithm’s effectiveness in optimizing route planning.

Discussion

The experimental results, supported by ANOVA analysis, demonstrate the effectiveness of the proposed hybrid path planning algorithms in improving the navigation performance and energy efficiency of AMRs in warehouse environments. The key findings and their implications are discussed below: Energy efficiency: The ANOVA results indicate that both the algorithm and map complexity have a significant effect on energy consumption (p ¡ 0.001), with a significant interaction effect as well. IACO+A*+DWA consistently demonstrated the lowest energy consumption across all map scenarios. For example, in the complex map scenario, IACO #x002B;A*+DWA consumed 89.8 J, which was 13.4% lower than the 103.7 J consumed by ACO+A*+DWA, 5.9% lower than the 95.4 J consumed by GA+A*+DWA, and 25.7% lower than the 120.9 J consumed by PSO+A*+DWA. This significant energy savings highlights the efficiency of IACO+A*+DWA, particularly in complex environments.

Path optimization: The ANOVA analysis also shows significant differences in path length between algorithms and map complexity, with a significant interaction effect. IACO+A*+DWA performed best in terms of minimizing path length and the number of turns. In the complex map scenario, IACO+A*+DWA resulted in a path length of 95.9 m with 43 turns, compared to 97.4 m with 46 turns for ACO+A*+DWA, 104.4 m with 50 turns for GA+A*+DWA, and 97.8 m with 56 turns for PSO+A*+DWA. IACO+A*+DWA reduced the path length by 1.5% compared to ACO+A*+DWA, 8.1% compared to GA+A*+DWA, and 1.9% compared to PSO+A*+DWA. The significant interaction effect (p ¡ 0.001) demonstrates that the algorithm’s performance is closely tied to the complexity of the map.

Time efficiency: The ANOVA results indicate that both the algorithm and map complexity significantly affect navigation time, with a significant interaction effect. IACO+A*+DWA completed the complex map scenario in an average of 186.0 s, which was 7.5% faster than ACO+A*+DWA’s 201.0 s and 2.3% faster than GA+A*+DWA’s 190.4 s, while being similar to PSO+A*+DWA (185.7 s). Although the time differences are relatively small, IACO+A*+DWA’s overall balance of energy consumption and path optimization makes it more effective.

Dynamic adaptability: The inclusion of DWA in the hybrid algorithms is crucial, allowing the robot to adjust its path dynamically in response to obstacles. The ANOVA results highlight the significant interaction effect of algorithm and map complexity on the number of turns, indicating that IACO+A*+DWA performs exceptionally well in dynamically adjusting the path, with fewer turns than the other algorithms. This confirms its suitability for real-world applications in dynamic environments like warehouses.

Overall performance comparison: The ANOVA analysis clearly highlights that IACO+A*+DWA outperforms the other algorithms across all performance metrics. The significant interaction effects of algorithm and map complexity on energy consumption, path length, number of turns, and navigation time indicate that IACO+A*+DWA is the most robust and efficient solution, especially in terms of energy efficiency and path optimization, regardless of map complexity.

Based on the ANOVA analysis and experimental results, the IACO+A*+DWA hybrid path planning algorithm consistently demonstrates superior performance in both complex and simple map environments, particularly in terms of energy efficiency and path optimization. Future research could further optimize the parameter settings of this algorithm, especially when tested in a wider variety of warehouse layouts. Additionally, integrating machine learning techniques could further enhance the algorithm’s adaptability, allowing it to dynamically learn and adjust to changing environments and obstacles, improving the robustness and flexibility of AMR navigation. As map complexity increases, IACO+A*+DWA’s advantages remain prominent. Future work could also focus on integrating this algorithm with warehouse management systems to achieve more efficient coordination between path planning and logistics, ultimately improving overall operational efficiency.

Conclusions

This research presented a novel hybrid path planning algorithm that integrates A*, ACO, and DWA to enhance the energy efficiency and navigation performance of AMRs in warehouse environments. The key contributions and findings of this research are summarized as follows: Hybrid algorithm development: A hybrid path planning algorithm was developed, combining the heuristic capabilities of A*, the optimization strength of ACO, and the dynamic obstacle avoidance of DWA. This integration allows the algorithm to optimize path planning in terms of energy consumption, path length, time efficiency, and adaptability in dynamic environments.

Performance evaluation: Extensive experiments were conducted across different warehouse scenarios, evaluating the proposed algorithms based on key performance metrics such as energy consumption, path length, number of turns, and time taken to reach the goal. The experimental results and ANOVA analysis revealed that the IACO+A*+DWA hybrid algorithm consistently outperformed other approaches across all metrics. It achieved the lowest energy consumption, shortest path length, fewer turns, and competitive navigation times, making it the most efficient algorithm tested.

Energy efficiency: IACO+A*+DWA demonstrated superior energy efficiency across all map complexities. For instance, in the complex map scenario, IACO+A*+DWA consumed only 89.8 J of energy, significantly lower than the 103.7 J consumed by ACO+A*+DWA, the 95.4 J consumed by GA+A*+DWA, and the 120.9 J consumed by PSO+A*+DWA. This result highlights the algorithm’s ability to optimize energy use while maintaining effective navigation performance.

Path optimization and time efficiency: The ANOVA results showed significant differences in path length and time efficiency, where IACO+A*+DWA demonstrated a 1.5% reduction in path length compared to ACO+A*+DWA and an 8.1% reduction compared to GA+A*+DWA. Additionally, the algorithm showed competitive navigation times, with a completion time of 186.0 s in the complex map scenario, outperforming GA+A*+DWA by 2.3%.

Dynamic obstacle avoidance: The inclusion of DWA in the hybrid algorithm was crucial for ensuring real-time adaptability to dynamic obstacles. This feature enables the robot to adjust its path in response to moving obstacles, making the IACO+A*+DWA algorithm particularly suitable for real-world applications in warehouse environments where dynamic obstacles are common.

Practical applicability: The proposed hybrid algorithms are practically applicable to real-world warehouse settings, offering a balanced solution that addresses the limitations of individual path-planning methods. The combination of energy efficiency, path optimization, and real-time obstacle avoidance makes the IACO+A*+DWA algorithm an ideal candidate for deployment in logistics and manufacturing sectors.

In conclusion, the hybrid path planning algorithm that integrates A*, ACO, and DWA provides a robust and energy-efficient solution for AMR navigation in warehouse environments. It demonstrates superior performance across various scenarios, particularly in energy consumption, path length, and adaptability to dynamic environments. Future work will focus on further optimizing the algorithm, exploring the integration of machine learning techniques to enhance real-time adaptability, and expanding the approach to different robotic systems and environments. The promising results of this study pave the way for more advanced and sustainable robotic solutions that can significantly impact the logistics and manufacturing industries, promoting greater efficiency and energy savings in automation.

Supplemental Information

Supplemental Information 1 Complete experimental code, including four algorithms.

Run the main3.m and main4.m files and wait to obtain the experimental results.

Supplemental Information 2 Map for Warehouse (complex map).

Supplemental Information 3 Warehouse map (basic map).

Additional Information and Declarations

Competing Interests

Author Contributions

Data Availability

The authors declare that they have no competing interests.

Mingyu Wu conceived and designed the experiments, performed the experiments, analyzed the data, performed the computation work, prepared figures and/or tables, authored or reviewed drafts of the article, and approved the final draft.

Eileen Lee Ming Su conceived and designed the experiments, authored or reviewed drafts of the article, and approved the final draft.

Che Fai Yeong conceived and designed the experiments, authored or reviewed drafts of the article, and approved the final draft.

Bowen Dong conceived and designed the experiments, performed the computation work, authored or reviewed drafts of the article, and approved the final draft.

William Holderbaum conceived and designed the experiments, authored or reviewed drafts of the article, and approved the final draft.

Chenguang Yang conceived and designed the experiments, authored or reviewed drafts of the article, and approved the final draft.

The following information was supplied regarding data availability:

The code is available in the Supplemental File.

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
