# Peer review of "A hybrid path planning algorithm combining A* and improved ant colony optimization with dynamic window approach for enhancing energy efficiency in warehouse environments"

_PeerJ Computer Science, doi:10.7717/peerj-cs.2629_

## Round 0.1 · original submission · Minor Revisions

The review of this paper has received positive comments, and the author is requested to make careful revisions according to the comments of the reviewers, further improve the paper and enhance the rigor of the paper.

Reviewer 1 ·

Basic reporting

the paper is professionally written, very clear and concise. relevant literatures were included and properly structured and formatted.

Experimental design

The proposed experiment is properly outlined and presented. the proposed method and robot energy model make sense and mathematically sound.

The research question whether combining well known optimization techniques such as Particle Swarm Optimization (PSO), Genetic Algorithm (GA), and Improved Ant Colony Optimization (IACO) with A* path finding algorithm and the Dynamic Window Approach (DWA) to dynamically adjust the path for energy efficient robot navigation is well defined.

The proposed experiment is properly outlined and presented. The proposed method and robot energy model make sense and mathematically sound.

Validity of the findings

acceptable (can be further enhanced with the disable of obstacle or multiple obstacles along the path)

The use of warehouse simulation in matlab by loading the map defined in excel file allowed the evaluation of robot' energy consumption, path length, number of turns, and time taken to reach the goal to be objectively measured.

Methods described with sufficient detail but would require the matlab code to be publicly available to ease the reproduction.

The contribution of the work is well concluded at the end of the paper.

Additional comments

good job, well done.

Reviewer 2 ·

Basic reporting

no comment

Experimental design

The hyperparameters of the models should be explicitly reported. For instance, for PSO, what C1 and C2 constants are used? What is the number of iterations used? The same goes for the other optimisation algorithms used. This would ensure the replicability of the work.

Validity of the findings

The readers would benefit from looking at the convergence curves for the evaluated cases (optional).

Additional comments

The authors present a novel hybrid path planning algorithm combining A*, Ant Colony Optimization (ACO), and the Dynamic Window Approach (DWA) to enhance energy efficiency in warehouse environments for autonomous mobile robots (AMRs). The authors compare their proposed IACO+A*+DWA approach against other hybrid methods combining A* with ACO, Genetic Algorithm (GA), and Particle Swarm Optimization (PSO).

The following are my suggestions:
1. Include a discussion on the computational complexity of the hybrid approach and how it scales with increasing map size and complexity.
2. Discuss potential limitations of the approach and scenarios where it might not perform as well.
3. Include a section on future work, such as incorporating learning algorithms to adapt to changing warehouse layouts or integrating with other warehouse management systems.

Reviewer 3 ·

Basic reporting

1.1. The manuscript is written in clear, professional English. However, minor grammatical improvements could enhance readability.
1.2. The introduction provides a good context and background, referencing relevant literature and establishing the importance of the research. However, the introduction could benefit from a more detailed explanation of the specific limitations of the current state-of-the-art algorithms.
1.3. The literature review section is comprehensive, covering ACO, GA, PSO, and their integration with A*. However, a table summarizing the advantages and disadvantages of these algorithms would improve readability and provide a quick reference for readers.
1.4. Raw data has been supplied, adhering to the journal's policy.
1.5. "Among the algorithms evaluated, A* demonstrated superior energy efficiency, particularly in complex environments." This could be clarified by specifying what makes A* particularly efficient in these environments.
1.6. Provide a more detailed explanation of the specific limitations of current state-of-the-art algorithms to better justify the need for the proposed hybrid approach.
1.7. Consider adding a table summarizing the advantages and disadvantages of the discussed algorithms (ACO, GA, PSO) for quick reference.
1.8. Clarify the mathematical equations with proper formatting and ensure consistency. For instance, ensure that all variables are defined immediately after their first use.
1.9. Ensure that all figures and tables are referenced in the text.

Experimental design

2.1. The research falls within the journal's scope and addresses a well-defined, relevant, and meaningful question. The hybrid algorithm aims to fill a knowledge gap in energy-efficient path planning in dynamic warehouse environments.
2.2. The investigation is performed to a high technical and ethical standard. The methods are described in sufficient detail to allow replication.
2.3. The methods section provides sufficient detail, including the mathematical models, experimental setup, and evaluation metrics, ensuring that the experiments can be replicated.
2.4. Improve the explanation of the integration of A*. The description can be more concise, focusing on the key steps and their significance.

Validity of the findings

3.1. The underlying data is robust, statistically sound, and controlled. The manuscript includes a detailed comparative analysis and statistical validation of the results.
3.2. The conclusions are well-stated and directly linked to the original research question. The findings support the claims made regarding the hybrid algorithm's superiority in terms of energy efficiency and path optimization.
3.3. Although the impact and novelty are not assessed, the manuscript demonstrates meaningful replication and provides a clear rationale and benefit to the literature.
3.4. When discussing the results, be more explicit about the statistical significance of the differences observed. For example, "The IACO+A*+DWA approach consistently demonstrated the lowest energy consumption across different map scenarios" could be strengthened by providing statistical evidence.

Additional comments

4.1. The manuscript presents an approach to path planning in warehouse environments, combining heuristic and optimization techniques effectively. The results are compelling, showing significant improvements in energy efficiency and path optimization.
4.2. Future research directions, such as integrating machine learning techniques, are mentioned, which adds value to the manuscript.
4.3. Ensure that all figures, tables, and equations are consistently formatted according to the journal's guidelines.
4.4. Check for any typographical errors and correct them.
4.5. Ensure all references are formatted correctly and consistently.

---

## Round 0.2 · Major Revisions

The authors are requested to revise the article in detail according to the comments of reviewer 3, and reply to all the comments raised by the reviewer.

Reviewer 2 ·

Basic reporting

No further comments - the authors have addressed this in the revised manuscript

Experimental design

No further comments - the authors have addressed this in the revised manuscript

Validity of the findings

No further comments - the authors have addressed this in the revised manuscript

Additional comments

No further comments - the authors have addressed this in the revised manuscript

Reviewer 3 ·

Basic reporting

The manuscript is written in clear, professional English and is generally easy to follow. However, some sentences could be more concise, and a few grammatical errors should be corrected (e.g. "An intelligence-based hybrid" in the abstract). The language is good quality, but I would benefit from additional proofreading and editing.

The manuscript provides a fairly thorough review of relevant literature on ACO, GA and PSO for robot path planning. Key studies are cited, and their findings are summarized to give context. However, the background and motivation for this particular study could be strengthened. Why is energy efficiency a critical issue for AMRs in warehouses specifically? Are there unique challenges or constraints in this environment that necessitate new path-planning approaches? Providing more context upfront would help convey the significance and novelty of this work. Some additional key references on energy-efficient path planning for AMRs seem to be missing. A deeper literature review could identify the remaining gaps this study aims to address.

The manuscript follows a standard structure with clearly delineated Abstract, Introduction, Methodology, Results and Discussion sections. The figures and tables are relevant and informative. However:
- Figure captions could be more descriptive to allow them to stand alone. For example, state what the coloured lines and shapes represent in Figure 3.
- Table 2 would be easier to read if presented in a graphical format instead, such as a bar chart comparing the metrics for each algorithm.
- Some key information appears to be missing, like the specific map dimensions and number of obstacles used in experiments. Including these details is important for reproducibility.

The study presents a new hybrid path-planning algorithm and evaluates it against several baselines. The methodology, results and major findings are presented in a self-contained way. However, some key information is lacking for the results to be fully reproduced and validated:
- The "Experimental Setup" section lacks details on the computing hardware used, software versions, and hyperparameter tuning process. More specifics are needed.
- Formal definitions of all performance metrics (energy consumption, time, etc.) and cost functions should be provided early on and referenced consistently. For example, how exactly is energy consumption calculated - based on distance, turns, time, or other factors?
- Proofs of optimality or convergence of the proposed IACO+A*+DWA hybrid approach are not provided. Theoretical analysis and justification of why this particular combination of algorithms is optimal could greatly strengthen the work.

Experimental design

Lack of statistical analysis: The results in Table 2 report raw values for each metric and algorithm, but no statistical comparisons are made. To draw robust conclusions, statistical significance tests (e.g. t-tests, ANOVA) should be used to determine if the differences between algorithms are meaningful or due to random chance. Reporting p-values and effect sizes is necessary.

No mention of multiple trials: It's unclear if the results reported are from a single run of each algorithm or averaged across multiple trials. For reliable results, each algorithm should be run multiple times on each map with different random seeds. Metrics should be averaged across trials, and standard deviations should be reported to assess variability.

Map details not provided: The "Experimental Setup" section does not specify the dimensions of the "basic" and "complex" maps, nor the number, size and placement of obstacles. These details are critical for interpreting results and comparing to other studies. Maps should be illustrated and quantitative details provided.

Narrow scope of environments tested: Only two simulated warehouse maps are used for evaluation. To demonstrate general applicability, the algorithms should ideally be tested on a wider range of environments, including other warehouse layouts and real-world settings, if possible. Simulations are a good start but have limitations.

Sensitivity analysis not performed: The study uses fixed parameter settings for each algorithm but does not investigate sensitivity to these choices. Parameters like pheromone weights, population size, probability constants, etc., can significantly impact performance. Sensitivity analysis should be performed to assess robustness.

Validity of the findings

The study could benefit from more rigorous statistical analysis. Including measures of statistical significance, confidence intervals, or effect sizes would provide a clearer picture of the reliability and magnitude of the observed differences between algorithms.

Conducting a sensitivity analysis on the algorithm parameters would help understand how robust the performance improvements are to changes in these settings.

Additional comments

While the research presents valuable contributions to robotic path planning in warehouse environments, there are a few key areas for enhancement. Firstly, a more in-depth discussion comparing your approach to other state-of-the-art methods in the field would provide valuable context. Secondly, elaborating on the practical implications of the energy efficiency improvements, such as potential cost savings or reduced environmental impact in real-world warehouse operations, would strengthen the paper's relevance. Lastly, while some limitations are mentioned, a dedicated section discussing the current study's limitations and how they might be addressed in future work would add depth to your analysis. Addressing these points could further enhance your manuscript's overall quality and impact.

---

## Round 0.3 · accepted · Accept

The author's modification of the paper has met the reviewer's requirements and improved the quality of the paper, so it is recommended to accept the paper.

Reviewer 3 ·

Basic reporting

The authors have comprehensively addressed the points of concern in their revised manuscript, warranting no further modifications.

Experimental design

The authors have comprehensively addressed the points of concern in their revised manuscript, warranting no further modifications.

Validity of the findings

The authors have comprehensively addressed the points of concern in their revised manuscript, warranting no further modifications.

Additional comments

The authors have comprehensively addressed the points of concern in their revised manuscript, warranting no further modifications